# Unlocking the power of synergy: High-intensity functional training and early time-restricted eating for transformative changes in body composition and cardiometabolic health in inactive women with obesity

Ranya Ameur[1,2☯], Rami Maaloul[1,3☯]*, Sémah Tagougui[4,5], Fadoua Neffati[6], Faten Hadj Kacem[7], Mohamed Fadhel Najjar[6], Achraf Ammar[3,8], Omar Hammouda[3,9]

1 High Institute of Sport and Physical Education of Sfax, University of Sfax, Sfax, Tunisia, 2 Research Laboratory of Evaluation and Management of Musculoskeletal System Pathologies, LR20ES09, University of Sfax, Sfax, Tunisia, 3 Research Laboratory, Molecular Bases of Human Pathology, LR19ES13, Faculty of Medicine, University of Sfax, Sfax, Tunisia, 4 Montreal Clinical Research Institute, Montreal, Canada, 5 University of Lille, University of Artois, University of Littoral Côte, d'Opale, ULR 7369-URePSSS-Multidisciplinary Research Unit, "Sport, Health and Society", Lille, France, 6 Biochemistry Laboratory, University Hospital of Monastir, Monastir, Tunisia, 7 Endocrinology Department, Hedi Chaker Hospital, University of Sfax, Sfax, Tunisia, 8 Department of Training and Movement Science, Institute of Sport Science, Johannes Gutenberg-University Mainz, Mainz, Germany, 9 Interdisciplinary Laboratory in Neurosciences, Physiology and Psychology: Physical Activity, Health and Learning (LINP2), UFR STAPS, UPL, Paris Nanterre, Nanterre, France

☯ These authors contributed equally to this work.
* rami.maaloulll@gmail.com

## Abstract

### Objective

The purpose of this study was to examine the long-term effects of time-restricted eating (TRE), with or without high intensity functional training (HIFT), on body composition and cardiometabolic biomarkers among inactive women with obesity.

### Methods

Sixty-four women (BMI = 35.03 ± 3.8 kg/m$^2$; age = 32.1 ± 10 years) were randomly allocated to either: (1) TRE (≤8-h daily eating window, with ad libitum energy intake) group; (2) HIFT (3 sessions/week) group; or (3) TRE combined with HIFT (TRE-HIFT) group. The interventions lasted 12 weeks with a pre-post measurement design. A HIFT session consists of 8 sets of multiple functional exercises with self-selected intensity (20 or 30s work/10s rest).

### Results

TRE-HIFT showed a greater decrease of waist and hip circumferences and fat mass compared to TRE (p = 0.02, p = 0.02 and p<0.01; respectively) and HIFT (p = 0.012, p = 0.028 and p<0.001; respectively). Weight and BMI decreased in TRE-HIFT compared to HIFT group (p<0.001; for both). Fat-free mass was lower in TRE compared to both HIFT and

**Data Availability Statement:** The data underlying the results presented in the study are available as supporting information files.

**Funding:** The authors received no specific funding for this work.

**Competing interests:** The authors have declared that no competing interests exist.

TRE-HIFT groups (p<0.01 and p<0.001; respectively). Total cholesterol, triglyceride, insulin, and HOMA-IR decreased in TRE-HIFT compared to both TRE (p<0.001, p<0.01, p = 0.015 and p<0.01; respectively) and HIFT (p<0.001, p = 0.02, p<0.01 and p<0.001; respectively) groups. Glucose level decreased in TRE-HIFT compared to HIFT (p<0.01). Systolic blood pressure decreased significantly in both TRE-HIFT and HIFT groups compared to TRE group (p = 0.04 and p = 0.02; respectively).

## Conclusion

In inactive women with obesity, combining TRE with HIFT can be a good strategy to induce superior effects on body composition, lipid profile and glucose regulation compared with either diet or exercise intervention alone.

## Trial registration

Clinical Trials Number: PACTR202301674821174.

## Introduction

Obesity is a critical clinical and public health issue as it is associated with an increased risk of cardiovascular disease, insulin resistance, cancer, oxidative stress and osteoarthritis [1]. One of the main approaches to combat obesity is by recommending lifestyle modifications such as changes in dietary behavior and increasing physical activity. These changes have shown promising results in weight reduction and maintenance in people with obesity [2]. In recent years, time-restricted eating (TRE) has gained attention as an intermittent fasting regimen that has the potential to be an alternative to continuous calorie restriction. TRE involves extending the fasting interval between the last evening meal and the first meal of the next day [3]. TRE has been demonstrated to produce similar improvements in cardiometabolic health (i.e., fat mass (FM), insulin sensitivity, glucose, lipid profile and blood pressure) as continuous calorie restriction diets [4]. This strategy has gained popularity since it is easy to follow for a longer period of time and does not require people to limit overall food consumption or calculate total daily calorie intake, which may improve adherence [3]. Calorie intake restricted to ≤8–10 hours per day has been proven effective in reducing obesity, inflammation, insulin resistance, and cardiometabolic dysfunction in both animal and human models [5, 6]. Short-term TRE has been investigated in several small clinical trials to study its weight-loss benefits in obese individuals. While some of these clinical trials [7–9] have shown TRE's weight reduction effects, not all have reported similar results [10]. A meta-analysis of 19 studies concluded that TRE significantly decreased body weight and fat mass while preserving fat-free mass (FFM) [11]. Furthermore, TRE has been shown to improve cardiometabolic biomarkers, including systolic blood pressure, fasting blood glucose, and triglycerides (TG), suggesting its usefulness in managing weight and reducing metabolic dysfunction in overweight and obese individuals [11].

Physical activity is another potent treatment in combating obesity [2], with High-Intensity Interval Training (HIIT) being recognized as an effective and time-efficient exercise for obese individuals [12]. Despite the benefits of HIIT, adherence to it remains suboptimal [13]. Cycling, running, and rowing are traditional exercise modalities that have adapted HIIT protocols, however for people exercising for health and leisure, these traditional modalities appear

boring and do not engage people due to the monotonous nature of the exercise associated with repetition [14]. High-Intensity Functional Training (HIFT), a newer exercise modality that combines aerobic and resistance exercises, has been proposed as an alternative to HIIT [15]. HIFT consists of alternating short periods of intense exercise with rest or active recovery periods of moderate intensity, using various multi-joint functional movements, performed at high relative intensities [15]. In overweight or obese adults with type 2 diabetes mellitus, HIFT has been shown to improve beta-cell function, reduce fat mass, and preserve lean mass [16]. Additionally, Fealy, et al. [17] reported improvements in cardiometabolic risk factors and insulin sensitivity after six weeks of HIFT. Research has shown that combining interventions (bidomain interventions) may generate better cardiometabolic adaptation in the obese population. However, there have only been two recent studies investigating the short-term synergistic effects of combining TRE and exercise training on body composition and cardiometabolic parameters in overweight/obese individuals [18, 19]. These studies reported that combining TRE with HIFT and HIIT, respectively, led to improved glycated hemoglobin, reductions in total and visceral fat mass, and greater weight loss. However, research on the long-term effects of TRE combined with training programs on body composition and cardiometabolic biomarkers is still lacking. Therefore, the aim of our study was to investigate the isolated and combined effects of 12 weeks of TRE and HIFT on body composition and cardiometabolic biomarkers in inactive obese women.

## Materials and methods

### Participants

A total of 64 women with obesity participated voluntarily in this study (age: 32.1 ± 10 years, body mass index (BMI): 35.03 ± 3.8 kg/m$^2$) (Table 1). Participants were recruited using social media advertisements, flyers, and word of mouth. The inclusion criteria were: (i) Females aged 18–45 years old (ii) BMI higher than 30 kg/m$^2$, (iii) waist circumferences (WC) higher than 80 cm, (iv) able to engage in physical activity, (v) had not followed a structured training program in the previous 6 months and, (vi) had maintained a constant weight for three months prior to the study. Women were not eligible for participation if they meet any of the following criteria: (i) pregnant, (ii) breastfeeding within 24 weeks of the start of the trial, (iii) having cardiovascular disease, (iv) having a diagnosis of diabetes, (v) using regular medications, (vi) smokers, (vii) who were undergoing weight loss treatment (viii) habitual eating window <12 hours/day.

**Table 1. Baseline characteristics of the participants.**

| Characteristics | TRE-HIFT (n = 20) | TRE (n = 20) | HIFT (n = 24) |
|---|---|---|---|
| Age (years) | 33.8 ± 5.3 | 30.2 ± 6.1 | 31.8 ± 2.2 |
| Weight (kg) | 96.2 ± 10.1 | 92.8 ± 14.2 | 89 ± 10.2 |
| Height (cm) | 164.7 ± 0.05 | 162.5 ± 0.05 | 159.8 ± 0.05 |
| Body mass index (kg/m$^2$) | 35.2 ± 3.5 | 35 ± 4.3 | 34.8 ± 3.6 |
| Waist circumference (cm) | 103.8 ± 7.7 | 98.9 ± 10 | 104 ± 8.7 |
| Hip circumference (cm) | 121.1 ± 5.5 | 160.3 ± 10.1 | 119.2 ± 5.2 |
| Waist-to-hip ratio | 0.85 ± 0.06 | 0.84 ± 0.09 | 0.87 ± 0.08 |
| Basal metabolism (kcal) | 1863.8 ± 425.9 | 1735.7 ± 215.2 | 1697.7 ± 369.4 |
| Resting Systolic blood pressure (mmHg) | 121.2 ± 6.5 | 121.7 ± 10.8 | 125.6 ± 10.8 |
| Resting diastolic blood pressure (mmHg) | 74.7 ± 5.7 | 73.5 ± 6.9 | 74.8 ± 5.8 |

Data are presented as mean ± standard deviation; **abbreviations**: TRE, time-restricted eating; HIFT, high intensity functional training

Before starting the study, all participants were informed about the experimental procedures, the possible risks, and discomforts associated with the study and signed a written informed consent. The study was carried out in accordance with the Helsinki Declaration and was approved by the "South Human Protection Ethics Committee" (C.P.P.SUD), Sfax, Tunisia: protocol reference C.P.P.SUD N° 0200/2019, prior to the beginning of the assessments (S1 Protocol). The study was registered at the database (PACTR202301674821174).

## Study design

Participants were assessed before and after completing 12 weeks of intervention for the three groups. All post-intervention measurements were carried out 48 hours following the last exercise session and/or the last day of the TRE protocol. The participants were randomized in a 1:1:1 manner to TRE (n = 20), HIFT (n = 24) and TRE-HIFT (n = 20) groups (Fig 1). The randomization of the participants was carried out using a web-based system for random number generation by a person independent of the research group. Participants arrived at the laboratory between 07:00 am and 09:00 am on two separate days before and after the intervention period to undertake assessment of body composition, fasting blood sample collection and blood pressure. Participants in the combined group (TRE-HIFT) followed the same exercise

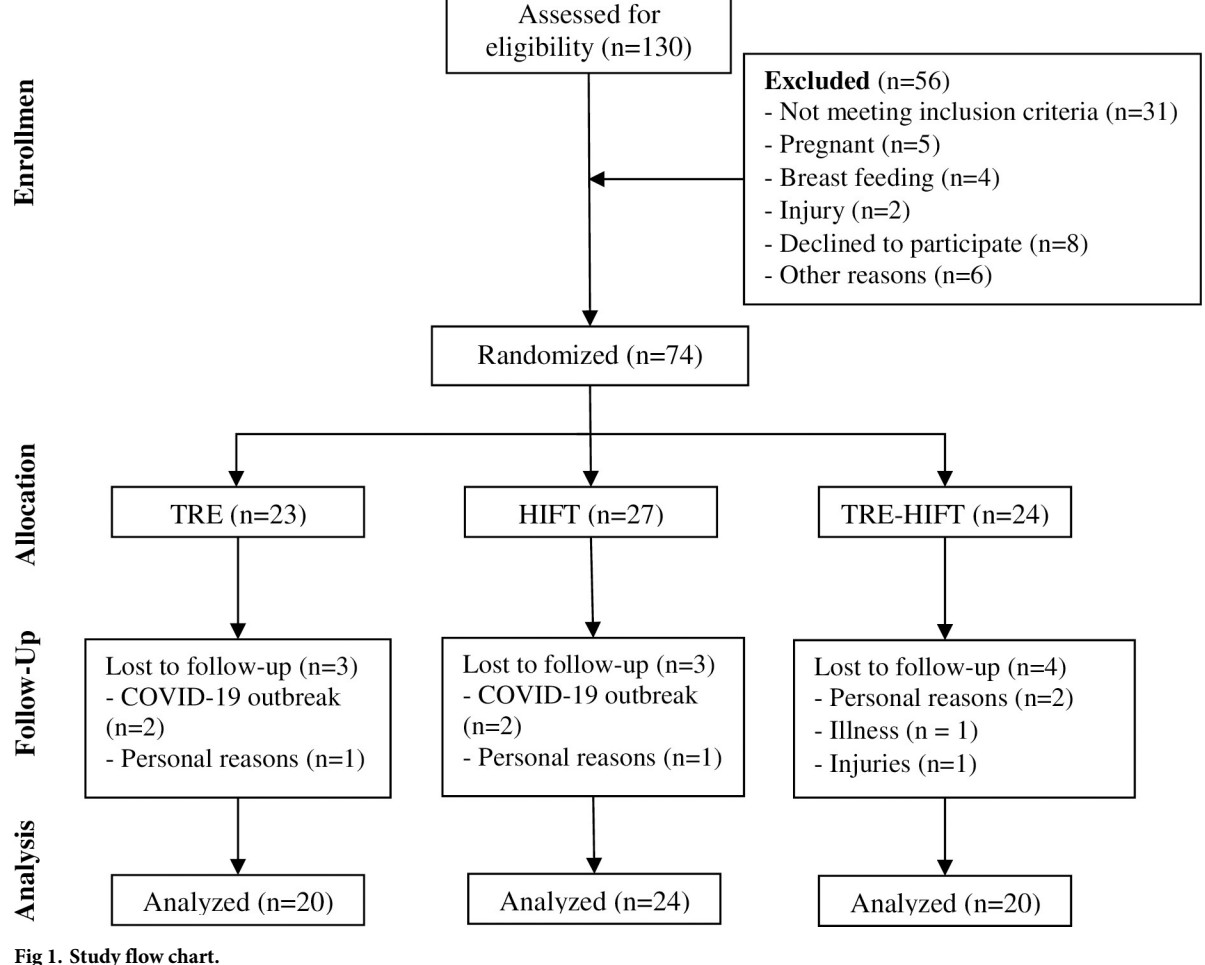

**Fig 1. Study flow chart.**

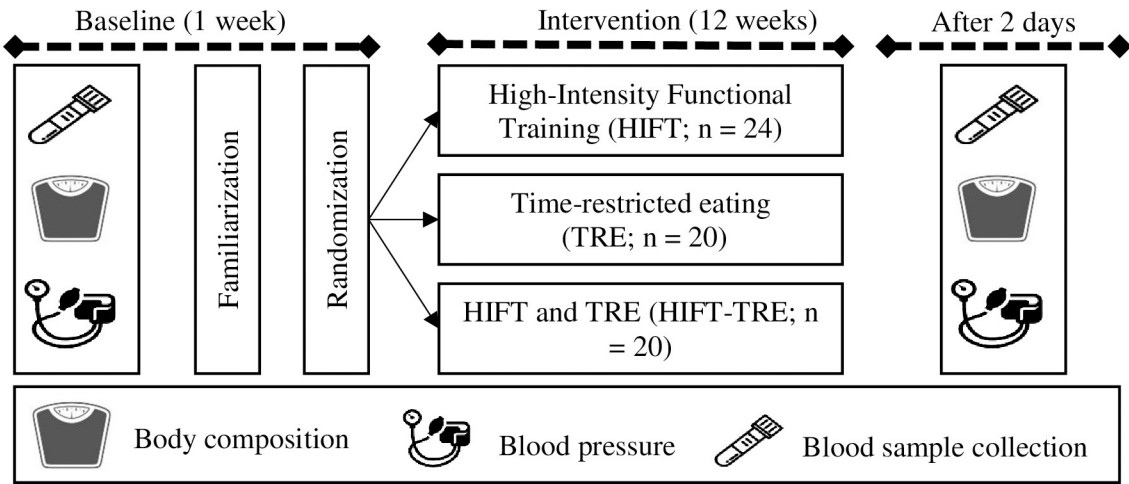

**Fig 2. Study design.**

protocol as per the HIFT group and the same TRE protocol as the TRE group (Fig 2) (S1 Checklist).

## Diet intervention

The TRE protocol adopted in this study for both TRE and TRE-HIFT groups was a fasting period of 16 h (from 4:00 pm to 8:00 am) and ad libitum eating $\leq$ 8 h (from 8:00 am to 4:00 pm). During the fasting period, participants will be allowed to consume water and calorie-free beverages. The participants of the HIFT group were required to maintain their regular eating schedule. Participants who followed the TRE diet received a daily reminder through instant messaging of the time to stop eating and the time when food was allowed. They were also asked about the problems or challenges encountered during the realization of the study protocol.

## HIFT program

One week prior to the 12 weeks of HIFT, two familiarization sessions were provided to acquaint the participants with the training procedure. HIFT was performed at 3 days per week on Monday, Wednesday, and Friday in the evening at the fasting window (i.e., 5:00 pm). All HIFT sessions were led by experienced instructors who followed similar exercise routines that included 5 minutes of dynamic warm-up and 5 minutes of cool-down, in addition to 45–55 minutes of HIFT. Each HIFT session consists of 8 sets of 8 functional exercises (aerobic and resistance) based on Tabata training [20]. All exercises were performed with body weight or with free weights (e.g., barbell, kettlebell, weight plate) using self-selected pace and weight/resistance. The Borg RPE CR10 Scale [21] was displayed to the participant repeatedly during exercise to maintain a minimum rating of perceived exertion (RPE) equal to 7. Participants were motivated to complete as many repetitions of a given exercise as possible over 20 s or 30 s followed by a 10 s rest. There was 2 min of rest period between each exercise. An example of an HIFT session is presented in Fig 3.

## Anthropometric and body composition assessment

Standing height was measured to the nearest 0.1 cm using a wall-mounted stadiometer. Height and weight were measured while participants were fasting and dressed in light clothing. Body

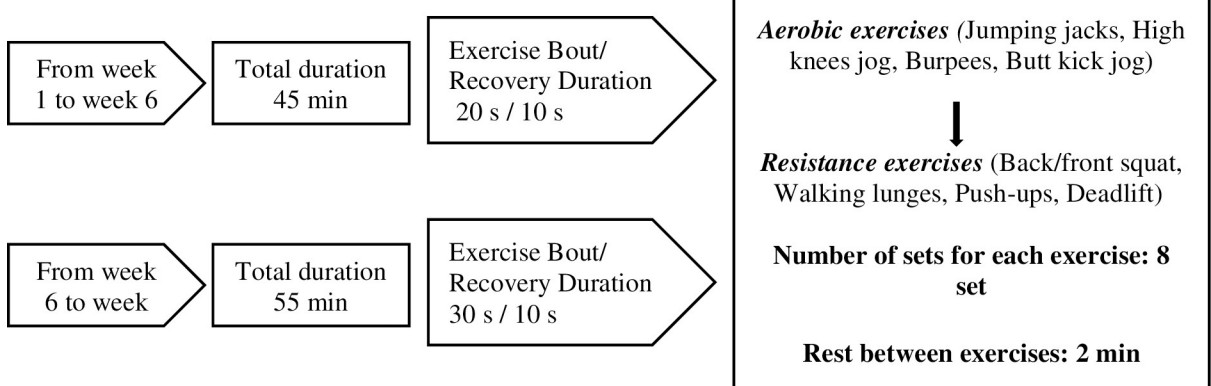

**Fig 3. Details of the high-intensity interval functional training program.**

composition was assessed using Tanita bioelectrical impedance (model TBF-300; Tanita Corp, Tokyo, Japan) and measurements of body weight, FFM and FM were recorded for each participant. Waist circumference (WC) was measured with a non-deformable tape ruler between the lower rib margin and the iliac crest, at the end gentle-expiration. Hip circumference (HC) was measured at the maximum protuberance of the buttocks. WC and HC were measured to the nearest 0.5 cm. Waist-to-hip ratio (WHR) was calculated as WC divided by HC.

## Blood pressure assessment

Blood pressure was measured before the blood test after 15 min rest in a seated position using a digital blood pressure monitor (Rossmax International LtD, Taipei, Taiwan). All measurements were assessed by the study physician who recorded systolic (SBP) and diastolic (DBP) blood pressures in mm Hg.

## Biochemical analysis

One week before and 2 days after the intervention, blood samples (10 ml) were taken from the antecubital vein at rest in the morning (between 7:00h and 9:00h) after an overnight fasting (~12 h). The blood collected was distributed in lithium heparin tube (5 ml) for glucose and lipid biomarkers assessment and an EDTA tube (5 ml) for insulin measurement, and immediately centrifuged at 2500 rpm for 10 min at 4˚C. The plasma was separated into aliquots and held at −80˚C until analyzed in the same day and in the same assay run to eliminate inter-assay-variance. Glucose level was measured with the glucose hexokinase method. Total cholesterol (TC), high-density lipoprotein (HDL) and TG were assessed using the Roche Cobas 6000 (c501) analyzer. Low-density lipoprotein (LDL) level have been estimated by the Friedewald equation: [LDL] = [TC]—([HDL] + [TG/2.2]) [22]. Insulin was measured with an immune-enzymatic method (CX9 Beckman coulter). Insulin resistance has been assessed using the homeostatic model assessment for insulin resistance (HOMA-IR). The HOMA-IR has been computed as follows: HOMA-IR = [insulinemia (µU/mL) × glycemia (mmol/L)] / 22.5 [23].

## Nutritional intake assessment

The nutritional data of the participants were collected by a nutritionist using a food diary for 7 days, one week before and at the last week of the intervention. All the participants received detailed verbal and written instructions on the process of recording their diet. A list of common household measures (e.g., tablespoons, cups) and specific information about the quantity

in each measurement (grams, etc.) was given to each participant. Each individual's diet was calculated using the Bilnut 2.01 software package (SCDA Nutrisoft, Cerelles, France) and the food composition tables published by the Tunisian National Institute of Statistics in 1978.

## Statistical analysis

SPSS statistical software (v.23, IBM, New York, USA) was used to analyze the data of the present study. The results were presented as mean ± standard deviation (SD) in tables. Shapiro–Wilk's W test was conducted to assess the normality and Levene's test was used to assess homogeneity of variances (signification was set at $p > 0.05$ for both tests). The paired t-test was employed to analyze within-group differences. Analysis of covariance (ANCOVA) was applied to compare variables following the intervention. In order to reduce the influence of within-group variability, we fixed as dependent variable the post value of the measured parameters for each group and the baseline values of the outcomes were adopted as a covariate. For multiple comparisons. Bonferroni post-hoc test was used. Partial eta squared effect sizes ($\eta p^2$) was used as an indicator of effect size of ANCOVA. Value of p lower than 0.05 was considered statistically significant.

## Results

A total of 130 women who responded to our request to participate were evaluated. 56 participants were withdrawn from the analysis (31 did not meet the selection criteria, 5 pregnant, 4 breast-feeding within 24 weeks of the start of the study, 8 declined to participate, 2 injured and 6 did not enter the study for other reasons). 74 women were actually enrolled in the study and randomized into TRE, HIFT and TRE-HIFT groups. 13.5% of participants who were allocated were unable to continue the training program or the post-intervention outcome measures. The final sample consisted of 64 women (TRE, n = 20; HIFT, n = 24; TRE-HIFT, n = 20).

### Body composition

After intervention, all groups (i.e., TRE, HIFT and TRE-HIFT) showed a significant decrease in body weight, BMI, WC, HC, WHR and FM ($p<0.001$; for all). FFM increased in HIFT and TRE-HIFT ($p<0.001$; for both), while it remained unchanged in the TRE group. A significant main effect of group by time was revealed for body weight ($p<0.01$, $\eta p^2 = 0.21$), BMI ($p<0.001$, $\eta p^2 = 0.22$), WC ($p<0.01$, $\eta p^2 = 0.15$), HC ($p = 0.017$, $\eta p^2 = 0.12$), FM ($p<0.001$, $\eta p^2 = 0.33$) and FFM ($p<0.001$; $\eta p^2 = 0.27$). In the between groups comparison, TRE-HIFT showed a greater decrease of WC ($p = 0.02$ and $p = 0.012$; respectively), HC ($p = 0.02$ and $p = 0.028$; respectively) and FM ($p<0.01$ and $p<0.001$; respectively) compared to TRE and HIFT groups. Also, a greater decrease of weight and BMI was observed in TRE-HIFT compared to HIFT group ($p<0.001$; for both). FFM was lower in TRE compared to HIFT and TRE-HIFT groups ($p<0.01$ and $p<0.001$; respectively). No significant differences were observed for WHR between groups (Table 2).

### Dietary intake

After 12 weeks of intervention, the TRE-HIFT, TRE, and HIFT groups showed a significant decrease in total calories intake ($p<0.001$; for all), carbohydrate ($p<0.001$, $p<0.01$ and $p<0.01$; respectively) and fat ($p<0.001$, $p<0.001$ and $p<0.01$; respectively). Protein intake decreased only in the TRE-HIFT group ($p<0.05$). A significant main effect of group by time was revealed for calories ($p <0.001$, $\eta p^2 = 0.25$) and fat ($p <0.001$, $\eta p^2 = 0.24$). In the between groups comparison, calorie intake decreased significantly in TRE-HIFT compared to the

**Table 2. Data related to body composition pre and post intervention.**

| Variable | Baseline mean ± SD | | | Post-intervention mean ± SD | | | Ancova | |
|---|---|---|---|---|---|---|---|---|
| | TRE-HIFT (n = 20) | TRE (n = 20) | HIFT (n = 24) | TRE-HIFT (n = 20) | TRE (n = 20) | HIFT (n = 24) | Intervention | Groups |
| **Weight** (kg) | 96.2 ± 10.1 | 92.8 ± 14.2 | 89 ± 10.2 | 84.6 ± 9.7*** | 83.8 ± 12.5*** | 83.6 ± 11.3*** | $F_{(1,60)} = 351.4$ $p < 0.001$; $\eta p^2 = 0.85$ | $F_{(2,60)} = 8$ $p < 0.01$; $\eta p^2 = 0.21$ |
| **BMI** (kg/m$^2$) | 35.2 ± 3.5 | 35 ± 4.3 | 34.8 ± 3.6 | 31 ± 3.3*** | 31.7 ± 3.9*** | 32.7 ± 4.1*** | $F_{(1,60)} = 254.4$ $p < 0.001$; $\eta p^2 = 0.8$ | $F_{(2,60)} = 8.7$ $p < 0.001$; $\eta p^2 = 0.22$ |
| **WC** (cm) | 103.8 ± 7.7 | 98.9 ± 10 | 104 ± 8.7 | 93.3 ± 5.6*** | 92.2 ± 9.3***,# | 96.4 ± 8.3***,# | $F_{(1,60)} = 303.2$ $p < 0.001$; $\eta p^2 = 0.83$ | $F_{(2,60)} = 5.6$ $p < 0.01$; $\eta p^2 = 0.15$ |
| **HC** (cm) | 121.1 ± 5.5 | 160.3 ± 10.1 | 119.2 ± 5.2 | 110.4 ± 5.3*** | 111.6 ± 8.6***,# | 113.2 ± 7.7***,# | $F_{(1,60)} = 66.9$ $p < 0.001$; $\eta p^2 = 0.52$ | $F_{(2,60)} = 4.4$ $p = 0.017$; $\eta p^2 = 0.12$ |
| **WHR** | 0.85 ± 0.06 | 0.84 ± 0.09 | 0.87 ± 0.08 | 0.84 ± 0.04 | 0.82 ± 0.09 | 0.85 ± 0.08 | $F_{(1,60)} = 94.8$ $p < 0.001$; $\eta p^2 = 0.61$ | $F_{(2,60)} = 0.1$ $p = 0.87$ |
| **FM** (kg) | 39.3 ± 9.2 | 35.6 ± 7.9 | 35.8 ± 8.7 | 31 ± 6.7*** | 31.6 ± 6.1***,## | 32.9 ± 8.6***,### | $F_{(1,60)} = 347.8$ $p < 0.001$; $\eta p^2 = 0.85$ | $F_{(2,60)} = 14.9$ $p < 0.001$; $\eta p^2 = 0.33$ |
| **FFM** (kg) | 23.3 ± 2.9 | 25.1 ± 3.7 | 22.1 ± 3.3 | 26.9 ± 3.2*** | 24.1 ± 3.41 | 25.6 ± 3.2***,### | $F_{(1,60)} = 37.9$ $p < 0.001$; $\eta p^2 = 0.38$ | $F_{(2,60)} = 11.3$ $p < 0.001$; $\eta p^2 = 0.27$ |

Data are presented as mean ± standard deviation; **abbreviations:** TRE, time-restricted eating; HIFT, high intensity functional training; BMI, body mass index; FM, fat mass; FFM, fat-free mass; WC, waist circumference; HC, hip circumference; WHR, Waist-to-hip ratio. ANCOVA was used for comparison between groups and the baseline values of the outcomes considered as covariate

***: significant change from baseline (p <0.001).

#, ##, ###: significant difference from TRE+HIIT group after intervention (p<0.05, p<0.01 and p<0.001; respectively).

HIFT group (p<0.001) and tended to be lower in TRE compared to HIFT group (p = 0.07). TRE-HIFT and TRE groups showed a greater decrease in Fat (g) compared to HIFT group (p<0.01; for both) (Table 3).

## Lipid biomarkers

TC decreased in TRE-HIFT, TRE and HIFT groups (p<0.001, p<0.001 and p<0.01; respectively). Similarly, LDL decreased in all groups (p<0.001) pre-post intervention. However, TG decreased only in TRE-HIFT and HIFT groups (p<0.01 and p<0.05; respectively). Moreover, HDL increased in TRE-HIFT, TRE and HIFT groups pre-post intervention (p<0.001, p<0.01 and p<0.001; respectively). A significant main effect of group by time was revealed for TC (p <0.01; $\eta p^2 = 0.2$) and TG (p <0.01; $\eta p^2 = 0.17$). No significant effect of group by time was observed for HDL and LDL. In the between groups comparison, TRE-HIFT noted a greater decrease in TC (p<0.001; for both) and TG (p<0.01 and p = 0.02; respectively) compared to TRE and HIFT groups (Table 4).

## Glucose, insulin and HOMA-IR biomarkers

TRE-HIFT, TRE and HIFT groups showed a significant decrease in glucose (p<0.001; for all), insulin (p<0.001, p<0.05 and p<0.01; respectively) and HOMA-IR (p<0.01; for all) pre-post intervention. A significant main effect of group by time was revealed for glucose (p <0.01, $\eta p^2 = 0.15$), insulin (p <0.01, $\eta p^2 = 0.16$) and HOMA-IR (p <0.01, $\eta p^2 = 0.22$). In the between groups comparison, TRE-HIFT noted a greater decrease of glucose (p<0.01) compared to HIFT and a greater decrease of insulin (p = 0.015 and p<0.01; respectively) and HOMA-IR (p<0.01 and p<0.001; respectively) compared to TRE and HIFT groups (Table 5).

**Table 3. Data related to dietary intake pre and post intervention.**

| Variable | Baseline mean ± SD | | | Post-intervention mean ± SD | | | Ancova | |
|---|---|---|---|---|---|---|---|---|
| | TRE-HIFT (n = 20) | TRE (n = 20) | HIFT (n = 24) | TRE-HIFT (n = 20) | TRE (n = 20) | HIFT (n = 24) | Intervention | Groups |
| **Calorie** (kcal) | 2710.2 ± 211.4 | 2519.2 ± 136.6 | 2682.1 ± 176.9 | 2311.9 ± 174.8*** | 2272.9 ± 131.6*** | 2461.1 ± 168.8***, ### | $F_{(1,60)} = 47.8$ $p$ <0.001 ; $\eta p^2 =$ 0.44 | $F_{(2,60)} = 10.4$ $p$ <0.001 ; $\eta p^2 =$ 0.25 |
| **Carbohydrate** (g) | 339 ± 31.4 | 306.4 ± 22.9 | 332.8 ± 26.5 | 294.6 ± 28.6*** | 281.6 ± 23.7** | 303 ± 33.4** | $F_{(1,60)} = 3.4$ $p = 0.07$ | $F_{(2,60)} = 1.4$ $p = 0.24$ |
| **Fat** (g) | 100 ± 8.5 | 95.6 ± 8.5 | 100.7 ± 11.2 | 83.8 ± 8.8*** | 83.7 ± 6.8*** | 92.8 ± 7.8** | $F_{(1,60)} = 11.3$ $p$ <0.01 ; $\eta p^2 = 0.15$ | $F_{(2,60)} = 9.5$ $p$ <0.001; $\eta p^2 =$ 0.24 |
| **Protein** (g) | 113.7 ± 26.6 | 108.3 ± 22.4 | 111.1 ± 10 | 94.9 ± 21.2* | 98.2 ± 21.1 | 103.4 ± 25.9 | $F_{(1,60)} = 0.8$ $p = 0.4$ | $F_{(2,60)} = 0.8$ $p = 0.45$ |
| **Carbohydrate** (%) | 50 ± 1.9 | 48.6 ± 2.6 | 49.7 ± 2.7 | 50.9 ± 2.8 | 49.6 ± 2.8 | 49.2 ± 3.7 | $F_{(1,60)} = 1.1$ $p = 0.3$ | $F_{(2,60)} = 2$ $p = 0.14$ |
| **Fat** (%) | 33.2 ± 2.3 | 34.1 ± 2.1 | 33.7 ± 2.6 | 32.6 ± 2.1*** | 33.1 ± 1.9*** | 33.9 ± 1.8***,## | $F_{(1,60)} = 0.1$ $p = 0.75$ | $F_{(2,60)} = 2.7$ $p = 0.08$ |
| **Protein** (%) | 16.7 ± 3.4 | 17.2 ± 3.4 | 16.6 ± 1.2 | 16.4 ± 3.6*** | 17.3 ± 3.6 | 16.8 ± 4.1*** | $F_{(1,60)} = 0.016$ $p = 0.9$ | $F_{(2,60)} = 0.2$ $p$ 0.78 |

Data are presented as mean ± standard deviation; **abbreviations:** TRE, time-restricted eating; HIFT, high intensity functional training. ANCOVA was used for comparison between groups and the baseline values of the outcomes considered as covariate

*, **, ***: significant change from baseline (p<0.05, p<0.01 and p<0.001; respectively).

##, ###: significant difference from TRE+HIIT group after intervention (p<0.01 and p<0.001; respectively).

## Blood pressure

TRE-HIFT and HIFT groups showed a significant decrease in SBP (p<0.001; for both) and DBP (p<0.01; for both) pre-post intervention. No significant difference from baseline was observed in TRE group. A significant main effect of group by time was revealed only for SBP (p = 0.011, $\eta p^2$ = 0.14). In the between groups comparison, SBP showed a greater decrease in both TRE-HIFT and HIFT groups compared to TRE group (p = 0.04 and p = 0.02; respectively) (Table 6).

**Table 4. Data related to lipid biomarkers pre and post intervention.**

| Variable | Baseline mean ± SD | | | Post-intervention mean ± SD | | | Ancova | |
|---|---|---|---|---|---|---|---|---|
| | TRE-HIFT (n = 20) | TRE (n = 20) | HIFT (n = 24) | TRE-HIFT (n = 20) | TRE (n = 20) | HIFT (n = 24) | Intervention | Groups |
| **TC** (mmol/L) | 5.2 ± 1.05 | 4.7 ± 0.97 | 5.1 ± 1 | 3.7 ± 0.42*** | 4.1 ± 0.84***, ### | 4.4 ± 0.97**, ### | $F_{(1,60)} = 32.6$ $p$ <0.001 ; $\eta p^2 = 0.35$ | $F_{(2,60)} = 7.5$ $p$ <0.01 ; $\eta p^2 = 0.2$ |
| **TG** (mmol/L) | 1.3 ± 0.69 | 1.3 ± 0.78 | 1.3 ± 0.43 | 0.8 ± 0.36** | 1.15 ± 0.7## | 1.1 ± 0.42*,## | $F_{(1,60)} = 50.2$ $p$ <0.001 ; $\eta p^2 = 0.45$ | $F_{(2,60)} = 6.4$ $p$ <0.01 ; $\eta p^2 = 0.17$ |
| **HDL** (mmol/L) | 1.2 ± 0.60 | 1.2 ± 0.33 | 1.4 ± 0.29 | 2.3 ± 0.38*** | 1.8 ± 0.61** | 2.1 ± 0.62*** | $F_{(1,60)} = 0.005$ $p = 0.94$ | $F_{(2,60)} = 2.9$ $p = 0.061$ |
| **LDL** (mmol/L) | 3.1 ± 0.82 | 3.1 ± 0.76 | 3.2 ± 0.92 | 1.6 ± 0.71*** | 2 ± 1.17*** | 1.9 ± 0.95*** | $F_{(1,60)} = 39.5$ $p$ <0.001 ; $\eta p^2 = 0.39$ | $F_{(2,60)} = 1.3$ $p = 0.27$ |

Data are presented as mean ± standard deviation; **abbreviations:** TRE, time-restricted eating; HIFT, high intensity functional training; TC, total cholesterol; TG, triglycerides; LDL, low-density lipoprotein; HDL, high-density lipoprotein. ANCOVA was used for comparison between groups and the baseline values of the outcomes considered as covariate.

*, **, ***: significant change from baseline (p<0.05, p<0.01 and p<0.001; respectively).

##, ###: significant difference from TRE+HIIT group after intervention (p<0.01 and p<0.001; respectively).

**Table 5. Data related to fasting blood glucose, plasma insulin and HOMA-IR pre and post intervention.**

| Variable | Baseline mean ± SD | | | Post-intervention mean ± SD | | | Ancova | |
|---|---|---|---|---|---|---|---|---|
| | TRE-HIFT (n = 20) | TRE (n = 20) | HIFT (n = 24) | TRE-HIFT (n = 20) | TRE (n = 20) | HIFT (n = 24) | Intervention | Groups |
| glucose (mmol/L) | 4.95 ± 0.75 | 4.72 ± 0.56 | 5.32 ± 0.67 | 3.72 ± 0.85*** | 3.76 ± 0.90*** | 4.57 ± 0.5***,## | $F_{(1,60)} = 13.45$ $p < 0.01$; $\eta p^2 = 0.18$ | $F_{(2,60)} = 5.28$ $p < 0.01$; $\eta p^2 = 0.15$ |
| Insulin (µUI/mL) | 13.51 ± 5.29 | 13.36 ± 4.35 | 13.29 ± 4.59 | 6.75 ± 3.2*** | 10.51 ± 4.56*,# | 10.58 ± 5.99**,## | $F_{(1,60)} = 21.65$ $p < 0.001$; $\eta p^2 = 0.26$ | $F_{(2,60)} = 6.02$ $p < 0.01$; $\eta p^2 = 0.16$ |
| HOMA-IR | 2.99 ± 1.33 | 2.8 ± 0.95 | 3.13 ± 1.15 | 1.08 ± 0.48*** | 1.79 ± 0.99***,## | 2.13 ± 1.19***,### | $F_{(1,60)} = 23.32$ $p < 0.001$; $\eta p^2 = 0.28$ | $F_{(2,60)} = 8.54$ $p < 0.01$; $\eta p^2 = 0.22$ |

Data are presented as mean ± standard error; **abbreviations:** TRE, time-restricted eating; HIFT, high intensity functional training; HOMA-IR, homeostasis model assessment-insulin resistance. ANCOVA was used for comparison between groups and the baseline values of the outcomes considered as covariate.

*, **, ***: significant change from baseline (p<0.05, p<0.01 and p<0.001; respectively).

#, ##, ###: significant difference from TRE+HIIT group after intervention (p<0.05, p<0.01 and p<0.001; respectively).

## Discussion

The purpose of this study was to examine the separate and combined effects of 12 weeks of early TRE (i.e., consuming all calorie intake between 8:00 a.m. and 4:00 p.m. daily) and HIFT (three exercise sessions per week), on body composition, cardiometabolic biomarkers, and calorie intake in women with obesity. The main findings of this study demonstrate that TRE combined with HIFT showed a synergistic effect on body composition and cardiometabolic health compared to either diet or exercise intervention. In addition, HIFT alone significantly improved blood pressure and FFM compared to TRE alone.

The present findings are consistent with previous studies that have examined the impact of TRE on changes in body composition. It has been reported that 8–16 weeks of TRE with different eating windows of 4–10 h was associated with a significant decrease in body weight by 3–4%, FM (i.e., relative and/or in percentage), and WC in people with overweight/obese [7, 8, 24] and in patients with metabolic syndrome independently of weight change [9]. Although in our study the TRE did not impose restrictions on total calorie intake or the macronutrient composition of foods, the weight loss may be related in part to the voluntary reduction in calorie intake that was observed in both TRE-HIFT (14.7%) and TRE groups (9.8%). Accordingly, it has been reported that individuals who followed this diet often spontaneously reduced their energy intake, resulting in a slight body weight loss of 1% to 4% over a period lasting from 1 week to 3 months [25].

**Table 6. Data related to blood pressure pre and post intervention.**

| Variable | Baseline mean ± SD | | | Post-intervention mean ± SD | | | Ancova | |
|---|---|---|---|---|---|---|---|---|
| | TRE-HIFT (n = 20) | TRE (n = 20) | HIFT (n = 24) | TRE-HIFT (n = 20) | TRE (n = 20) | HIFT (n = 24) | Intervention | Groups |
| SBP (mmHg) | 121.2 ± 6.5 | 121.7 ± 10.8 | 125.6 ± 10.8 | 115.5 ± 7.8*** | 120.7 ± 8.1# | 117.7 ± 9.1*** | $F_{(1,60)} = 55.1$ $p < 0.001$; $\eta p^2 = 0.48$ | $F_{(2,60)} = 4.9$ $p = 0.011$; $\eta p^2 = 0.14$ |
| DBP (mmHg) | 74.7 ± 5.7 | 73.5 ± 6.9 | 74.8 ± 5.8 | 71.5 ± 4.6** | 72.7 ± 5.9 | 71.9 ± 4.1** | $F_{(1,60)} = 18.8$ $p < 0.001$; $\eta p^2 = 0.23$ | $F_{(2,60)} = 0.9$ $p = 0.4$ |

Data are presented as mean ± standard deviation; **abbreviations:** TRE, time-restricted eating; HIFT, high intensity functional training; SBP, systolic blood pressure; DBP, diastolic blood pressure. ANCOVA was used for comparison between groups and the baseline values of the outcomes considered as covariate.

**, ***: significant change from baseline (p<0.01 and p<0.001; respectively).

#: significant difference from TRE+HIIT group after intervention (p<0.05).

Interestingly, in the current study, FFM did not decrease in the TRE group, indicating that TRE preferentially reduces FM without reducing muscle mass, which was consistent with a previous study [24]. This finding may help to explain why TRE could improve metabolic dysfunction in obese people, as muscle mass plays a crucial role for resting metabolism, glucose regulation, and maintaining skeletal integrity [26]. Given that other studies related to TRE have revealed a significant reduction in muscle mass [7, 8, 10]. The combination of an intermittent fasting regimen and regular exercise has recently emerged as a recommended strategy to mitigate the risk of unhealthy lean mass loss, and to improve the functional physical capacities of inactive adults with obesity during the weight-loss process [19, 27, 28]. In the present study, the combination TRE-HIFT showed a significant decrease in body weight, FM, HC and WC and an increase in FFM compared to the TRE or HIFT intervention alone. Similar findings have been reported in a four-week trial that compared Ramadan intermittent fasting (participants were allowed to eat between 7:00 p.m. and 3:00 a.m.) to a combined condition of Ramadan fasting with concurrent training (HIIT + resistance exercise) [28]. Results reported that the combined strategy induced greater decreases in weight, FM, Fat percentage, and WC, while FFM showed no change compared to the diet strategy alone [28]. Moreover, Kotarsky et al. [19] showed that 8 weeks of late TRE (eating window between 12:00 a.m. and 8:00 p.m.) associated with concurrent training showed a greater decrease in FM in comparison with the control group (i.e., resistance training alone). However, the FFM increased significantly in the control group and remained unchanged in the TRE group [19]. The disparities in FFM outcomes shown in these studies may be related to the various types of diet followed by participants, the duration of the training programs, and/or differences in the types of exercises.

Regarding exercise strategies that can promote muscle mass development, HIFT with self-selected intensity using constantly varied high-intensity functional and muscle-strengthening exercises has been proven to be useful as an alternative to HIIT running to eliminate exercise barriers for physical exertion and to increase strength, FFM, and exercise adherence [15, 29, 30]. Additionally, the present study revealed that lipid profiles (i.e., TG and TC), glucose, insulin, and HOMA-IR of participants undergoing TRE-HIFT improved more significantly than those of participants undergoing TRE or HIFT alone. The combined effects of fasting and physical training on cardiometabolic biomarkers in overweight/obese people are equivocal, with some studies demonstrating greater improvements in lipid profile, glucose regulation and blood pressure compared to the fasting or exercise intervention alone [18, 27, 28] while others have shown no synergistic effects [31]. For example, Ramadan diurnal intermittent fasting combined with concurrent training provided superior benefits for weight loss and greater improvements in cardiometabolic biomarkers (i.e., TC, TG and LDL) compared to either intervention alone [28]. Nevertheless, Cooke et al. [31] showed that twice weekly intermittent fasting with or without a sprint exercise training program (3 sessions/week) provided similar effects on body composition (i.e., FM and FFM). These authors reported no significant changes between groups for both peripheral and central blood pressure measurements, lipid biomarkers (i.e., TC and HDL), glucose, HbA1c, and HOMA-IR after 16 weeks of intervention in overweight and obese people. While glucose tolerance and LDL cholesterol were decreased in the combined group only [31]. When TRE and HIFT are used concurrently, a greater energy deficit is likely to be generated than when doing either strategy separately. As a result, the larger decrease in FM seen in the TRE-HIFT group may potentially be attributable to a beneficial metabolic switch between glucose oxidation and fat oxidation as glycogen stores are depleted [18, 32]. Indeed, in the fasting state, a metabolic shift from lipid synthesis and fat storage to fat mobilization has been triggered, and the body changes its source of energy from glucose to ketones through free fatty acid oxidation, which helps promote both muscle mass and function [33, 34]. Also, fasting induces AMPK phosphorylation, which activates genes

involved in regulating mitochondrial biogenesis and substrate utilization [34]. Similarly, those metabolic adaptations that occur while fasting are further activated in response to exercise in the fasted state [32, 35]. Exercise performed in fasted state has been demonstrated to increase free fatty acid mobilization and activate signaling pathways in the skeletal muscle, such as AMPK, in comparison to exercise performed in fed state [32, 35]. Additionally, it has been shown that exercise done while fasting can enhance fat oxidation at rest from 9 to 24 hours following exercise when compared to exercise performed when fed [32]. This higher utilization of fat as an energy source at rest may promote a reduction in FM.

The significant decrease in fasting glucose and HOMA-IR observed in this study may be explained by the similar processes by which TRE and HIT improve glucose control. Indeed, intermittent fasting approaches [33], similar to exercise [36], have been shown to stimulate mitochondrial biogenesis through the activation of AMPK and subsequent stimulation of PCG-1 alpha, as well as increasing GLUT4 translocation [33, 35, 36]. Additionally, previous studies have proven the effectiveness of HIFT in improving beta cell function as well as increasing insulin sensitivity in people with type 2 diabetes [16, 17]. Thus, the combination of TRE and HIFT may have synergistic effects that improve blood glucose regulation. Otherwise, the decrease in SBP and DBP in the TRE-HIFT and HIFT groups compared to the TRE group may highlight the benefits of adopting HIFT program to improve cardiometabolic health. Fealy et al. [17] reported significant decreases in DBP and lipid biomarkers (i.e., TG and VLDL) after 6 weeks of HIFT. The lack of changes in blood pressure observed in the TRE group may be due to the normal baseline values of the participants included in the present study (SBP = $121.7 \pm 10.8$ and DBP = $73.5 \pm 6.9$ mmHg).

In summary, combining TRE with HIFT can be a good strategy to induce superior effects on body composition, lipid profile, and glucose regulation compared to either diet or exercise intervention alone. However, HIFT would be required to improve FFM and cardiovascular health. Future TRE research should determine which time of exercising (i.e. fast/fed state) is more relevant for improving cardiometabolic health in women with obesity.

This study has several limitations. First, our sample size was small and not calculated and our study is most likely underpowered. Second, the HIFT program included a variety of functional exercises; therefore, the outcomes may differ if HIFT was applied with other combinations of movements. Third, we cannot determine whether the benefits of TRE on metabolic health are due to a reduced food window and/or a possible decrease in energy intake since calorie intake was decreased in both TRE and TRE-HIFT groups. In addition, food intake assessed by dietary intake logs can lead to inaccurate estimates of nutrient intake, as faults in the use of self-reported food intake diaries are well recognized, which is a limitation of using such tools. Finally, variations in menstrual cycles were not accounted for due to the longitudinal nature of the study; therefore, it was impossible to completely rule out any impacts that may influence our results.

## Supporting information

**S1 Checklist. CONSORT checklist.**
(DOCX)

**S1 Protocol. Study protocol.**
(DOCX)

## Acknowledgments

We thank the dieticians and biologists involved in the study for their cooperation and involvement with the study. We also thank the participants for sharing their views and experiences.

## Author Contributions

**Conceptualization:** Omar Hammouda.

**Data curation:** Ranya Ameur.

**Formal analysis:** Rami Maaloul.

**Investigation:** Ranya Ameur.

**Methodology:** Omar Hammouda.

**Project administration:** Omar Hammouda.

**Resources:** Fadoua Neffati, Faten Hadj Kacem, Mohamed Fadhel Najjar.

**Supervision:** Fadoua Neffati, Mohamed Fadhel Najjar.

**Validation:** Sémah Tagougui, Achraf Ammar.

**Visualization:** Rami Maaloul.

**Writing – original draft:** Ranya Ameur, Rami Maaloul.

**Writing – review & editing:** Sémah Tagougui, Achraf Ammar, Omar Hammouda.

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
