## [Decision Letter · Decision Letter 0]

27 Jul 2023

PONE-D-23-14396Unlocking the power of synergy: High-intensity functional training and early time-restricted eating for transformative changes in body composition and cardiometabolic health in inactive women with obesityPLOS ONE

Dear Dr. Maaloul,

Thank you for submitting your manuscript to PLOS ONE. After careful consideration, we feel that it has merit but does not fully meet PLOS ONE’s publication criteria as it currently stands. Therefore, we invite you to submit a revised version of the manuscript that addresses the points raised during the review process.

We look forward to receiving your revised manuscript.

Kind regards,

Niels Wedderkopp, Ph.D.

Academic Editor

PLOS ONE

Journal Requirements:

Additional Editor Comments:

We are sorry it has taken so long, but it was not possible to find reviewers enough. I would advice you to closely follow the reviewers advice and suggestions, then the manuscript could reach the quality needed to be published in PLOS ONE.

Reviewers' comments:

Reviewer's Responses to Questions

**Comments to the Author**

1. Is the manuscript technically sound, and do the data support the conclusions?

Reviewer #1: Yes

Reviewer #2: Partly

2. Has the statistical analysis been performed appropriately and rigorously? 

Reviewer #1: I Don't Know

Reviewer #2: No

3. Have the authors made all data underlying the findings in their manuscript fully available?

Reviewer #1: Yes

Reviewer #2: No

4. Is the manuscript presented in an intelligible fashion and written in standard English?

Reviewer #1: Yes

Reviewer #2: Yes

5. Review Comments to the Author

Reviewer #1: Abstract:

- "Cardiometabolic biomarkers (i.e., total cholesterol, triglyceride, insulin and HOMA-IR) improved significantly in TRE-HIFT compared with either intervention alone (p<0.05).", an expression that is prone to misinterpretation, as stand-alone groups have also produced effects. Please examine the whole text carefully; there are many more of this expression, so please consider revising it. It is not limited to the following places: "... showed a significant decrease in body weight, FM, HC and WC and an increase in FFM compared to the TRE or HIFT intervention alone." and "... undergoing TRE-HIFT improved more significantly than those of participants undergoing TRE or HIFT alone".

Introduction

- The authors argue that HIIT is boring and that HIFT will compensate for this deficiency. But the literature cited by the authors does not seem to support this. For example, "..., these traditional modalities appear boring and do not engage people due to the monotonous nature of the exercise associated with repetition [14]." and "Compared to HIIT, HIFT has been shown to have higher exercise adherence rates, with participant-controlled exercise intensity promoting better training compliance [18-20]."

- The authors may have overlooked that there is a large body of literature that supports the idea that HIIT has a better subjective perception.

- Also, in the methods section it is stated that HIFT lasts 45-55 minutes, which is clearly too long compared to many types of exercise in HIIT, which is also a disadvantage.

Methods

- There is a lack of monitoring data for HIFT training, such as heart rate and RPE, which is not conducive to replicating this form of exercise in future operator practice.

Discussion

- "Given that other studies [7, 8, 10] have revealed a..." Should it be stated here what specific studies are being conducted, simply TRE?

- "Feally et al. [17] reported significant..." , To logically connect with the next sentence, the subject's pre-intervention blood pressure status should be stated here.

Reviewer #2: The manuscript could be improved based on the following comments.

Baseline characteristics of the participants are to be presented and usually presented in the first table as Table 1.0

For the statistical analysis, the statement ‘both tests were considered significant for p> 0.05’ requires revision.

There was no information on sample size calculation provided.

Missing data (if any) is to be reported.

The statement ‘Bonferroni post-hoc test was also used for comparing intergroup changes when a significant interaction was detected’ is unclear and requires revision. If the interaction was explored, it should be mentioned explicitly in the results section. Effect size Cohen was mentioned but was not used or presented in the results section. Partial eta squared is to be mentioned in the statistical analysis section for ANCOVA. The type of adjustment/correction used in ANCOVA is to be clearly stated. Any statistical approach mentioned in the results section is to be highlighted in the statistical analysis section and vice versa. Anything that was not used/employed is to be removed.

The reason why ANCOVA was employed compared to other statistical tests such as MANCOVA and the fulfillment of the statistical test assumption is to be stated.

Sensitivity analysis could be explored.

Table 1, WHR was not discussed in the results section.

For the Dietary intake section, the page where Table 2 was presented, Line 3-4, there was an error in the p-value cited for the statement ‘TRE-HIFT and TRE groups showed a greater decrease in fat intake compared to HIFT group (p<0.01; for both) (table 2)’. Also, to state variable Fat(g) since there are two fat (g & %) and for between-group comparison in the text.

Table 1, decimal points are to be standardized where applicable.

For the Blood pressure section, for the significant main effect of group by time, the data/findings of intervention were mentioned instead of groups.

If the groups’ information in the last column of Tables 1, 2, 3, and 4 were mentioned, the same approach should be used to mention the intervention information (second last column).

Baseline adjustment is to be denoted in the footnote of tables.

n to be included in the tables and Figure 4.

Although the detailed results analysis was not presented for the glucose, insulin, and HOMA-IR biomarkers section in table form, rechecking the results is required to avoid misquoting in the text.

The results presentation especially the text results could be summarized rather than providing/citing too many figures which can be observed from the tables.

Information on who performed the randomization, allocation concealment, blinding (if any) is to be clearly stated.

Ensure whatever that is checked in the CONSORT checklist, the information is to be provided/mentioned in the manuscript and to comply the checklist parameters wherever possible.

For the study protocol, only the relevant content is to be attached with this manuscript.

The list of references did not comply with the journal format.

6. PLOS authors have the option to publish the peer review history of their article (what does this mean?). If published, this will include your full peer review and any attached files.

Reviewer #1: No

Reviewer #2: No

---

## [Author Response · Author response to Decision Letter 0]

6 Feb 2024

Point by point responses to the reviewers’ comments

Unlocking the power of synergy: High-intensity functional training and early time-restricted eating for transformative changes in body composition and cardiometabolic health in inactive women with obesity

We would like to thank the reviewers for their thoughtful review of the manuscript. They raise important issues and their suggestions are very helpful for improving the manuscript. We almost agree with all their comments and we have revised our manuscript accordingly.

We are already crafting a revised version of the paper that it states the hypothesis and the implications of our work more clearly than before. Moreover, we are including all reviewers’ suggestions and clarifying the text when needed in red color.

We respond below in detail to each of the reviewer’s comments. We hope that the reviewers will find our responses to their comments satisfactory.

Please, find below the reviewers’ comments repeated in italics and our responses inserted after each comment in red color. To facilitate the work of the reviewers, in some instances the references are written following each answer. 

Looking forward hearing from you soon.

Sincerely,

Rami Maaloul et al.

Reviewer's Responses to Questions

Comments to the Author

Review Comments to the Author

Reviewer 

#1: Abstract:

- "Cardiometabolic biomarkers (i.e., total cholesterol, triglyceride, insulin and HOMA-IR) improved significantly in TRE-HIFT compared with either intervention alone (p<0.05).", an expression that is prone to misinterpretation, as stand-alone groups have also produced effects. Please examine the whole text carefully; there are many more of this expression, so please consider revising it. It is not limited to the following places: "... showed a significant decrease in body weight, FM, HC and WC and an increase in FFM compared to the TRE or HIFT intervention alone." and "... undergoing TRE-HIFT improved more significantly than those of participants undergoing TRE or HIFT alone".

We kindly appreciate the comment of the reviewer; we did some modifications in the whole manuscript to clarify all the confusing sentences.

Results: 

TRE-HIFT showed a greater decrease of waist and hip circumferences and fat mass compared to TRE (p=0.02, p=0.02 and p<0.01; respectively) and HIFT (p=0.012, p=0.028 and p<0.001; respectively). Weight and BMI decreased in TRE-HIFT compared to HIFT group (p<0.001; for both). Fat-free mass was lower in TRE compared to both HIFT and TRE-HIFT groups (p<0.01 and p<0.001; respectively). Total cholesterol, triglyceride, insulin, and HOMA-IR decreased in TRE-HIFT compared to both TRE (p<0.001, p<0.01, p=0.015 and p<0.01; respectively) and HIFT (p<0.001, p=0.02, p<0.01 and p<0.001; respectively) groups. Glucose decreased in TRE-HIFT compared to HIFT (p<0.01). Systolic blood pressure decreased significantly in both TRE-HIFT and HIFT groups compared to TRE group (p=0.04 and p=0.02; respectively). 

Introduction

- The authors argue that HIIT is boring and that HIFT will compensate for this deficiency. But the literature cited by the authors does not seem to support this. For example, "..., these traditional modalities appear boring and do not engage people due to the monotonous nature of the exercise associated with repetition [14]." and "Compared to HIIT, HIFT has been shown to have higher exercise adherence rates, with participant-controlled exercise intensity promoting better training compliance [18-20]."

- The authors may have overlooked that there is a large body of literature that supports the idea that HIIT has a better subjective perception.

We kindly appreciate the comment of the reviewer. 

We totally agree with the Reviewer that HIIT is largely better than moderate intensity continuous training in term of training adherence and many other physiological outcomes. However, in the present study, we focused mainly on HIFT which seems even more motivating than HIIT, at least for the present study participants. Of note, the comparison between the two exercise modalities is not at the center of our study. We changed the reference number [14], and we deleted the second sentence "Compared to HIIT, HIFT has been shown to have higher exercise adherence rates, with participant-controlled exercise intensity promoting better training compliance [18-20]." Please see the changes in the corrected version.

Wilke J. Functional high-intensity exercise is more effective in acutely increasing working memory than aerobic walking: An exploratory randomized, controlled trial. Scientific Reports. 2020;10(1):12335.

- Also, in the methods section it is stated that HIFT lasts 45-55 minutes, which is clearly too long compared to many types of exercise in HIIT, which is also a disadvantage.

We totally agree with the reviewer. A typical HIIT workout usually lasts anywhere from 10-30 minutes. However, this exercise modality was chosen because HIFT can provide stronger motivation and higher exercise enjoyment than moderate aerobic type activity. Moreover, HIFT protocol used in this study corresponds to a program that combines aerobic and resistance exercises into one session (4 exercises; 8 sets x 20 s, 10 s set), resulting in an exercise duration of 16 min for each training component with 2 minutes of rest between exercises.

 Methods

- There is a lack of monitoring data for HIFT training, such as heart rate and RPE, which is not conducive to replicating this form of exercise in future operator practice.

We thank the reviewer for this comment. workload was monitored throughout the exercise and was based on a self-selected pace for an RPE ≥ 7 using the Borg RPE CR10 Scale. Unfortunately, it was not possible to control heart rate in this study.

Discussion

- "Given that other studies [7, 8, 10] have revealed a..." Should it be stated here what specific studies are being conducted, simply TRE?

We thank the Reviewer for his valuable comment, we indicated that these studies were conducted on the TRE diet and we reworded the paragraph as follow “ Given that other studies related to TRE have revealed a significant reduction in muscle mass [7, 8, 10]. The combination of an intermittent fasting regimen and regular exercise ….”

- "Feally et al. [17] reported significant..." , To logically connect with the next sentence, the subject's pre-intervention blood pressure status should be stated here.

We thank the reviewer for this comment, the subject's pre-intervention blood pressure status was added to the paragraph as follow. “Feally et al. [17] reported significant decreases in DBP and lipid biomarkers (i.e., TG and VLDL) after 6 weeks of HIFT. The lack of changes in blood pressure observed in the TRE group may be due to the normal baseline values of the participants included in the present study (SBP = 121.7 ± 10.8 and DBP = 73.5 ± 6.9 mmHg).”

Reviewer #2: The manuscript could be improved based on the following comments.

Baseline characteristics of the participants are to be presented and usually presented in the first table as Table 1.0

 Done, please see the changes in the corrected version. 

Table 1. Baseline characteristics of the participants 

Characteristics TRE-HIFT (n=20) TRE (n=20) HIFT (n=24) 

Age (years) 33.8 ± 5.3 30.2 ± 6.1 31.8 ± 2.2 

Weight (kg) 96.2 ± 10.1 92.8 ± 14.2 89 ± 10.2 

Height (cm) 164.7 ± 0.05 162.5 ± 0.05 159.8 ± 0.05 

Body mass index (kg/m²) 35.2 ± 3.5 35 ± 4.3 34.8 ± 3.6 

Waist circumference (cm) 103.8 ± 7.7 98.9 ± 10 104 ± 8.7 

Hip circumference (cm) 121.1 ± 5.5 160.3 ± 10.1 119.2 ± 5.2 

Waist-to-hip ratio 0.85 ± 0.06 0.84 ± 0.09 0.87 ± 0.08 

Basal metabolism (kcal) 1863.8 ± 425.9 1735.7 ± 215.2 1697.7 ± 369.4 

Resting Systolic blood pressure (mmHg) 121.2 ± 6.5 121.7 ± 10.8 125.6 ± 10.8 

Resting diastolic blood pressure (mmHg) 74.7 ± 5.7 73.5 ± 6.9 74.8 ± 5.8 

Data are presented as mean ± standard deviation; abbreviations: TRE, time restricted eating; HIFT, high intensity functional training

-For the statistical analysis, the statement ‘both tests were considered significant for p> 0.05’ requires revision.

We thank the reviewer for this comment, the sentence become “Shapiro–Wilk’s W test was conducted to assess the normality and Levene's test was used to assess homogeneity of variances (signification was set at p> 0.05 for both tests). ”

-There was no information on sample size calculation provided

Regrettably, sample size was not calculated a priori in this study. This lacking information is added as a limitation for the present study

 -Missing data (if any) is to be reported.

We thank the reviewer for this comment. There are no missing data for the present study results. All the participants were tested before and after the intervention.

-The statement ‘Bonferroni post-hoc test was also used for comparing intergroup changes when a significant interaction was detected’ is unclear and requires revision. If the interaction was explored, it should be mentioned explicitly in the results section. Effect size Cohen was mentioned but was not used or presented in the results section. Partial eta squared is to be mentioned in the statistical analysis section for ANCOVA. The type of adjustment/correction used in ANCOVA is to be clearly stated. Any statistical approach mentioned in the results section is to be highlighted in the statistical analysis section and vice versa. Anything that was not used/employed is to be removed.

-Sensitivity analysis could be explored.

We thank the reviewer for all these valuable comments, we reworded the statistical analysis section differently. “Shapiro–Wilk’s W test was conducted to assess the normality and Levene's test was used to assess homogeneity of variances (signification was set at p> 0.05 for both tests). The paired t-test was employed to analyze within-group differences. Analysis of covariance (ANCOVA) was applied to compare variables following the intervention. In order to reduce the influence of within-group variability, we fixed as dependent variable the post value of the measured parameters for each group and the baseline values of the outcomes were adopted as a covariate. For multiple comparisons, Bonferroni post-hoc test was used. Partial eta squared effect sizes (ηp2) was used as an indicator of effect size for ANCOVA. Value of p < 0.05 was considered statistically significant. »

-Table 1, WHR was not discussed in the results section.

We thank the reviewer for this comment, we discussed the results of WHR as follows “After intervention, all groups (i.e., TRE, HIFT and TRE-HIFT) showed a significant decrease in body weight, BMI, WC, HC, WHR and FM (p<0.001; for all)….. No significant differences were observed for WHR between groups (table 1).” 

-For the Dietary intake section, the page where Table 2 was presented, Line 3-4, there was an error in the p-value cited for the statement ‘TRE-HIFT and TRE groups showed a greater decrease in fat intake compared to HIFT group (p<0.01; for both) (table 2)’. Also, to state variable Fat(g) since there are two fat (g & %) and for between-group comparison in the text.

We thank the reviewer for this comment, the p value cited in the table (p<0.001) is the result of the ANCOVA test (group effect) however the value cited in the text “(p<0.01; for both)” is the result of the post-hoc test. We added Fat (g) in the corrected version.

-Table 1, decimal points are to be standardized where applicable.

Done, please see the changes in the corrected version

-For the Blood pressure section, for the significant main effect of group by time, the data/findings of intervention were mentioned instead of groups.

We kindly appreciate the comment of the reviewer, we are sorry for this mistake and we reworded the paragraph as following “TRE-HIFT and HIFT groups showed a significant decrease in SBP (p<0.001; for both) and DBP (p<0.01; for both) pre-post intervention. No significant difference from baseline was observed in TRE group. A significant main effect of group by time was revealed only for SBP (p =0.011, ηp2 = 0.14). In the between groups comparison, SBP showed a greater decrease in both TRE-HIFT and HIFT groups compared to TRE group (p=0.04 and p=0.02; respectively) (table 4).”

-If the groups’ information in the last column of Tables 1, 2, 3, and 4 were mentioned, the same approach should be used to mention the intervention information (second last column).

We thank the reviewer for this comment, the changes over time were discussed by comparing the results before and after the intervention. we presented only the group effect to avoid redundancy in the presentation of the results.

-Baseline adjustment is to be denoted in the footnote of tables.

Done, please see the changes in the corrected version. 

-n to be included in the tables and Figure 4.

Done, please see the changes in the corrected version. 

-Although the detailed results analysis was not presented for the glucose, insulin, and HOMA-IR biomarkers section in table form, rechecking the results is required to avoid misquoting in the text.

-The results presentation especially the text results could be summarized rather than providing/citing too many figures which can be observed from the tables.

We thank the reviewer for this comment, we deleted figure 4 and we presented the results of glucose, insulin, and HOMA-IR biomarkers in table form, please see table 5 

Table 5. Data related to fasting blood Glucose, plasma insulin and HOMA-IR pre and post intervention

Variable Baseline mean ± SD Post-intervention mean ± SD Ancova

 TRE-HIFT (n=20) TRE (n=20) HIFT (n=24) TRE-HIFT (n=20) TRE (n=20) HIFT (n=24) Intervention Groups

glucose (mmol/L) 4.95 ± 0.75 4.72 ± 0.56 5.32 ± 0.67 3.72 ± 0.85*** 3.76 ± 0.90*** 4.57 ± 0.5***,## F(1,60) = 13.45 p <0.01 ; ηp2 = 0.18 F(2,60) = 5.28 p <0.01 ; ηp2 = 0.15

Insulin (µUI/mL) 13.51 ± 5.29 13.36 ± 4.35 13.29 ± 4.59 6.75 ± 3.2*** 10.51 ± 4.56*,# 10.58 ± 5.99**,## F(1,60) = 21.65 p <0.001 ; ηp2 = 0.26 F(2,60) = 6.02 p <0.01 ; ηp2 = 0.16

HOMA-IR 2.99 ± 1.33 2.8 ± 0.95 3.13 ± 1.15 1.08 ± 0.48*** 1.79 ± 0.99***,## 2.13 ± 1.19***,### F(1,60) = 23.32 p <0.001 ; ηp2 = 0.28 F(2,60) = 8.54 p <0.01 ; ηp2 = 0.22

Data are presented as mean ± standard error; abbreviations: TRE, time restricted eating; HIFT, high intensity functional training; HOMA-IR, homeostasis model assessment-insulin resistance. ANCOVA was used for comparison between groups and the baseline values of the outcomes considered as covariate.

*, **, ***: significant change from baseline (p<0.05, p<0.01 and p<0.001; respectively).

#, ##, ###: significant difference from TRE+HIIT group after intervention (p<0.05, p<0.01 and p<0.001; respectively).

-Information on who performed the randomization, allocation concealment, blinding (if any) is to be clearly stated.

Done, please see the changes in the corrected version. “The participants were randomized in a 1:1:1 manner to TRE (n = 20), HIFT (n = 24) and TRE-HIFT (n = 20) groups (fig 1). The randomization of the participants was carried out using a web-based system for random number generation by a person independent of the research group.”

-Ensure whatever that is checked in the CONSORT checklist, the information is to be provided/mentioned in the manuscript and to comply the checklist parameters wherever possible.

We ensured that everything specified in the CONSORT checklist matched all the information in the manuscript.

-The list of references did not comply with the journal format.

Done, please see the changes in the corrected version.

---

## [Decision Letter · Decision Letter 1]

14 Mar 2024

Unlocking the power of synergy: High-intensity functional training and early time-restricted eating for transformative changes in body composition and cardiometabolic health in inactive women with obesity

PONE-D-23-14396R1

Dear Dr. Maaloul,

We’re pleased to inform you that your manuscript has been judged scientifically suitable for publication and will be formally accepted for publication once it meets all outstanding technical requirements.

Kind regards,

Niels Wedderkopp, Ph.D.

Academic Editor

PLOS ONE

Reviewers' comments:

Reviewer's Responses to Questions

**Comments to the Author**

1. If the authors have adequately addressed your comments raised in a previous round of review and you feel that this manuscript is now acceptable for publication, you may indicate that here to bypass the “Comments to the Author” section, enter your conflict of interest statement in the “Confidential to Editor” section, and submit your "Accept" recommendation.

Reviewer #1: All comments have been addressed

Reviewer #2: (No Response)

2. Is the manuscript technically sound, and do the data support the conclusions?

Reviewer #1: Yes

Reviewer #2: Partly

3. Has the statistical analysis been performed appropriately and rigorously? 

Reviewer #1: I Don't Know

Reviewer #2: Yes

4. Have the authors made all data underlying the findings in their manuscript fully available?

Reviewer #1: Yes

Reviewer #2: Yes

5. Is the manuscript presented in an intelligible fashion and written in standard English?

Reviewer #1: Yes

Reviewer #2: Yes

6. Review Comments to the Author

Reviewer #1: (No Response)

Reviewer #2: (No Response)

---

## [Editor Report · Acceptance letter]

8 Apr 2024

PONE-D-23-14396R1 

PLOS ONE

Dear Dr. Maaloul, 

I'm pleased to inform you that your manuscript has been deemed suitable for publication in PLOS ONE. Congratulations! Your manuscript is now being handed over to our production team.

Kind regards, 

on behalf of

Professor Niels Wedderkopp 

Academic Editor

PLOS ONE